# The Importance of Including Maternal Immune Activation in Animal Models of Hypoxic–Ischemic Encephalopathy

**DOI:** 10.3390/biomedicines12112559

**Published:** 2024-11-08

**Authors:** Bailey Collins, Elise A. Lemanski, Elizabeth Wright-Jin

**Affiliations:** 1Division of Biomedical Research, Nemours Children’s Health, Wilmington, DE 19803, USA; bailey.collins@nemours.org (B.C.); elise.lemanski@nemours.org (E.A.L.); 2Psychological and Brain Sciences, University of Delaware, Newark, DE 19716, USA; 3Division of Neurology, Nemours Children’s Health, Wilmington, DE 19803, USA; 4Sidney Kimmel Medical College, Thomas Jefferson University, Philadelphia, PA 19107, USA

**Keywords:** hypoxic–ischemic encephalopathy, brain injury, neuroinflammation, maternal immune activation, animal models

## Abstract

Hypoxic–ischemic encephalopathy (HIE) is a perinatal brain injury that is the leading cause of cerebral palsy, developmental delay, and poor cognitive outcomes in children born at term, occurring in about 1.5 out of 1000 births. The only proven therapy for HIE is therapeutic hypothermia. However, despite this treatment, many children ultimately suffer disability, brain injury, and even death. Barriers to implementation including late diagnosis and lack of resources also lead to poorer outcomes. This demonstrates a critical need for additional treatments for HIE, and to facilitate this, we need translational models that accurately reflect risk factors and interactions present in HIE. Maternal or amniotic infection is a significant risk factor and possible cause of HIE in humans. Maternal immune activation (MIA) is a well-established model of maternal infection and inflammation that has significant developmental consequences largely characterized within the context of neurodevelopmental disorders such as autism spectrum disorder and schizophrenia. MIA can also lead to long-lasting changes within the neuroimmune system, which lead to compounding negative outcomes following a second insult. This supports the importance of understanding the interaction of maternal inflammation and hypoxic–ischemic outcomes. Animal models have been invaluable to understanding the pathophysiology of this injury and to the development of therapeutic hypothermia. However, each model system has its own limitations. Large animal models such as pigs may more accurately represent the brain and organ development and complexity in humans, while rodent models are more cost-effective and offer more possible molecular techniques. Recent studies have utilized MIA or direct inflammation prior to HIE insult. Investigators should thoughtfully consider the risk factors they wish to include in their HIE animal models. In the incorporation of MIA, investigators should consider the type, timing, and dose of the inflammatory stimulus, as well as the timing, severity, and type of hypoxic insult. Using a variety of animal models that incorporate the maternal–placental–fetal system of inflammation will most likely lead to a more robust understanding of the mechanisms of this injury that can guide future clinical decisions and therapies.

## 1. Introduction

Hypoxic–ischemic encephalopathy (HIE) is a perinatal brain injury characterized by a lack of oxygen and blood flow to the fetal brain. Its causes can be attributed to a wide array of intrauterine and peripartum events including chorioamnionitis, maternal hypoxia, maternal hypotension, placental abruption, cord prolapse and uterine rupture, among many others [1]. The outcomes of this early-life injury vary widely, ranging from normal development and cognition to severe disability and static encephalopathy [2,3,4]. Children with severe HIE have a mortality rate of 25–50%, and those who survive can have significant long-term deficits including cerebral palsy, epilepsy, learning deficits, and vision/hearing impairments [5,6]. It was originally thought that children who did not experience physical disabilities such as cerebral palsy were not at risk for other significant neurodevelopmental disabilities. However, more recent research suggests that in children with no motor deficits following HIE, there can be brain injury that is detectable through abnormal MRI and cognitive impairments detectable only in later childhood [7,8,9].

There are mixed data regarding sex differences in incidence of HIE and severity of outcomes after HIE, with some studies reporting greater incidence and severity for boys [10,11,12,13,14] and others reporting no difference [15]. For preterm infants, increased negative outcomes for males is well documented [10,11,16], with males also being overrepresented in spontaneous preterm births and complications as a whole [16,17,18]. In a study following 308 infants born at less than 25 weeks gestation, males were found to be more likely to develop cerebral palsy and impaired cognitive function than their female counterparts [11]. The mechanisms behind this sex difference are unknown but are currently hypothesized to be due to differences in microglia activation, reactions to oxidative stress, and cell death between males and females [19]. Sex differences in prevalence and outcomes for term infants are less clear, though some studies reported being male was a risk factor for HIE [12]. Much of the research conducted on sex differences in term infants has been conducted using animal models of HIE [20], and there is a need for more studies to fill in this gap in the literature [14].

The neural injury following a hypoxic–ischemic event occurs in two waves, referred to as primary and secondary energy failure. Primary energy failure occurs immediately following the decrease in blood flow and oxygen to the brain. The lack of oxygen and glucose decreases available energy and forces cells to switch to anaerobic metabolism, resulting in a decrease in ATP production, lactate buildup, and sodium/potassium pump failure [21]. The buildup of intracellular sodium leads to chronic depolarization and the buildup of intracellular calcium [22]. The buildup of free radicals such as oxidative stress also contributes to injury [23]. The compounded cellular stress during primary energy failure can lead to cell death through necrosis [23]. Following reperfusion of blood to the brain, there is a brief period of recovery referred to as the latency period, the length of which depends on the severity of injury [24]. Secondary energy failure occurs following the latency period, approximately 6–48 h following the hypoxic–ischemic insult [24]. Excitotoxicity due to the influx of sodium in primary energy failure as well as oxidative stress continue into secondary energy failure [25]. Additionally, mitochondrial dysfunction, cytotoxic edema, and inflammatory activation such as cytokine storms begin to contribute to injury during this time [26,27,28]. Apoptosis (programmed cell death) is heightened during secondary energy failure [29].

Currently, the only effective treatment for HIE is therapeutic hypothermia (TH) [8,9]. This therapy involves cooling the head and body of the infant to 33.5 °C for 72 h. This decrease in temperature slows cerebral metabolism, preserves glucose reserves in the brain, and inhibits the buildup of molecules such as lactate [30]. TH also decreases free radicals and dampens the neuroimmune response [30]. TH significantly improves neurological outcomes in HIE, leading to decreased rates of mortality, severe developmental delay, cerebral palsy, cognitive delay, and severe motor impairment [24]. However, there are some limitations to TH as a treatment. TH must be instituted within the first 6 h after birth to be effective [31]. Intrapartum hypoxia can be difficult to diagnose as there is no definitive diagnostic test for this condition [32]. Neonatal encephalopathy is typically diagnosed through clinical presentations of symptoms such as the Apgar score, abnormal fetal heart rate, or need for respiratory support [24]. The severity of encephalopathy is then characterized by the Sarnat score and can be further categorized by neuroimaging [33]. The possible overlooking of clinical symptoms, along with other barriers to care, can make starting treatment within those first 6 h difficult. Even with successful implementation of therapeutic hypothermia, up to 40% of children suffer from long-term neurodevelopmental disability [6,34]. Additionally, in low-income countries, TH has been shown to be ineffective or harmful due to increased bleeding and thrombocytopenia [35]. This results in there being no effective treatments for infants with HIE in these countries, which comprise the greatest worldwide burden of disease. The need for more therapies for HIE is critical, and to develop them, a robust and comprehensive understanding of the pathophysiology of HIE is needed.

Animal models have greatly contributed to our understanding of the pathophysiology underlying HIE injury and later outcomes. However, following TH, no treatments developed in animal models have been shown to be effective in humans. This points to the need to re-evaluate the translational relevancy of the animal models used to develop these treatments. HIE poses a unique challenge due to the high variability in risk factors, causes, location and severity of injury, and functional outcomes that are present in humans [36]. Many animal models of HIE utilize postnatal animals and lack the maternal–placental–fetal interface that largely characterizes HIE [37]. Some additionally include procedures such as carotid artery ligation that produce unilateral lesions not characteristic of HIE injury in humans [37]. Prenatal risk factors and causes should be considered for inclusion within animal models. Maternal immune activation (MIA) is a significant risk factor for HIE and can in fact directly cause hypoxic injury [38,39]. MIA is easy to include in animal models through the peripheral injection of bacterial or viral mimetics into pregnant animals, thereby incorporating maternal and fetal interaction [40]. An additional benefit of MIA is the expansive literature characterizing the molecular, immune, and behavioral outcomes, largely in rodent models [40,41,42]. The purpose of this review is to discuss the importance and relevancy of MIA, summarize the current animal models of HIE, and discuss how animal models of HIE can incorporate MIA to facilitate more translationally relevant models of HIE.

## 2. Maternal Immune Activation Is an Epidemiological Risk Factor for Hypoxic–Ischemic Encephalopathy

Maternal immune activation (MIA) is an environmental factor that is a significant risk factor for the development of HIE and cerebral palsy [43,44,45,46]. Chorioamnionitis diagnosis via clinical symptoms or histological analysis of the placenta was found to lead to a 4-fold increased risk of cerebral palsy in term infants [47]. In one study of 587 infants who were diagnosed with cerebral palsy, maternal infection preceding delivery was present as a risk factor in 41.4% of children compared to controls [48]. A systematic review of risk factors for cerebral palsy in developed countries identified chorioamnionitis as a significant risk factor for the development of this disability [44]. Furthermore, forty percent of children diagnosed with HIE without a sentinel birth event had been exposed to infection when examined [49,50]. Of 137 infants diagnosed with HIE who did not experience a sentinel event, histological evidence of chorioamnionitis was more likely to be seen in placental pathology that demonstrated increased polymorphonuclear leukocytes [49]. Another study found that 40% of infants presenting with HIE had chorioamnionitis alone, and 11% had both chorioamnionitis and another sentinel birth event [50]. Intrapartum fever was associated with a 3.1-fold increased risk of encephalopathy, and chorioamnionitis was associated with a 5.4-fold increased risk of encephalopathy [38]. In another study, for both preterm and term deliveries, infection at the time of delivery was associated with a twofold increased risk of developing cerebral palsy [51]. This epidemiological evidence points to the need for the inclusion of factors such as MIA into our understanding of HIE pathology.

## 3. Maternal Immune Activation Has Been Well Characterized in Neurodevelopmental Disorders

MIA is a well-studied risk factor in neurodevelopment, as maternal infections have been clinically associated with the development of neuropsychiatric disorders such as autism spectrum disorder (ASD) and schizophrenia, as well as the development of epilepsy [52]. There appears to be a male bias in these disorders, as ASD has a higher occurrence in boys than girls at a rate of 3:1 [53], and schizophrenia is slightly more common in males [54]. However, incidence rates of epilepsy are similar between males and females [55]. Large cohort studies investigating risk factors for childhood seizures and epilepsy have found increased risk due to overall maternal infection, and more specifically to influenza, maternal cystitis, pyelonephritis, persistent diarrhea, cough, and vaginal yeast infections [56,57]. Multiple types of maternal infections have been epidemiologically associated with increased risk of ASD and schizophrenia. The link between influenza and schizophrenia was initially observed following the 1918–1919 Spanish influenza epidemic [58]; however, additional cohort studies have found mixed results on the strength of this association [59,60]. Large cohort studies for other infections have found that maternal infection with toxoplasma gondii or bacterial genitourinary infections are risk factors for schizophrenia [61], with mixed evidence for herpes simplex 2 (HSV-2) and Cytomegalovirus (CMV) [62,63]. Meta-analyses of clinical studies have found associations between ASD and any maternal infection, with increased risk in patients who required hospitalization for these infections [64]. Outside of discrete infections, increased expression of inflammatory markers such as cytokines or C-reactive protein have also demonstrated increased risk for both ASD and schizophrenia diagnosis [65,66,67,68]. Given that children affected by HIE may have a wide array of neurodevelopmental and psychiatric impairments following injury, this evidence suggests that the significant disruptions in neurodevelopment that accompany MIA and HIE may not be specific to these disorders [69,70].

In rodent models, MIA leads to a milieu of long-lasting behavioral changes representative of the symptomology of neurodevelopmental disorders. Social behavior deficits have been characterized in MIA models, including a decrease in play behavior and ultrasonic vocalizations and an increase in self-grooming behavior [71,72]. Repetitive behaviors such as those seen in ASD are characterized in rodent models through the marble burying task, which is increased in MIA animals [73,74]. MIA disrupts working memory and latent inhibition post-puberty, replicating well-characterized cognitive deficits in humans with schizophrenia [75,76,77,78]. Prepulse inhibition (PPI) is a behavioral task used to assess sensorimotor gating, a core phenotype of schizophrenia, and in which MIA animals exhibit deficits in adulthood [79,80,81]. There are critical windows of neurodevelopment in which MIA is correlated with specific phenotypes [82]. In rodents, earlier gestational exposure (E9 to E12) to MIA has been shown to induce sensorimotor gating deficits and differences in reward-seeking behavior, whereas later gestational exposure (E16 to E18) impaired learning and memory as well as caused motor deficits [83,84]. Rodents exposed to MIA have also exhibited sleep disturbances and increased susceptibility to seizures [85,86,87,88]. Many of the negative outcomes following MIA are worse in males, which is consistent with humans, as neurodevelopmental disorders occur more often in boys [89]. This suggests that there may be a differential function of the immune system within development for males and females [90].

## 4. Maternal Immune Activation Disrupts Neurodevelopment Through Maternal and Fetal Inflammation

Rodent models are the most commonly used method to study the mechanisms of MIA. These models typically utilize peripheral injections of proinflammatory molecules such as lipopolysaccharide (LPS) or polyriboinosinic–polyribocytidylic acid (Poly I:C) at critical points during gestation. LPS is a bacterial cell wall component derived from Gram-negative bacteria that functions as an endotoxin and binds to toll-like receptor 4 (TLR4) to stimulate the release of cytokines and other immune molecules (Figure 1). There are two major downstream pathways following LPS binding: the MyD88-dependent pathway and the MyD88-independent pathway [91,92,93]. In the MyD88-dependent pathway, LPS binding triggers signaling via MYD88, TRIF, and IRAK4, IRAK2, and IRAK1 to ultimately initiate activation of transcription factors like NF-κB to trigger an inflammatory cytokine response [91]. The MyD88-independent pathway is also known as the TRIF-dependent pathway [92]. In this signaling pathway, LPS binding to TLR4 triggers TIR-domain-containing adaptor protein inducing IFN-β (TRIF) and TRIF-related adaptor molecule (TRAM) [94]. This triggers IRAF4, TRAF6, then c-Rel to stimulate IL-12 expression [93]. Poly(I:C) is a synthetic analogue of a double-stranded viral RNA that binds to toll-like receptor 3 (TLR3), triggering an immune reaction similar to a viral infection [94] (Figure 1). TLR3 is an intracellular receptor located on the endoplasmic reticulum, endosomes, and lysosomes that double-stranded RNA bind to and activate [95]. It is the only member of the receptor family solely dependent on TRIF signaling [96]. Upon binding, TRIF begins signaling cascades that lead to IRF3 and NF-κB activation [97]. The IRF3 pathway begins when TRIF recruits TNF receptor associated factor 3 (TRAF3), which then associates with TBK1 and IKKi. The TBK1 and IKKε complex then phosphorylates IRF3, which then translocates to the nucleus to trigger an inflammatory response, particularly those mediated via type 1 interferons [98]. TRIF can also activate NF-κB via activating the IKK complex, which will lead to increased cytokine production [97].

The immune response of the pregnant dam has traditionally been thought to be the main cause of neural and developmental changes within the fetus. Following the binding to toll-like receptors by pathogens or mimetics and the activation of proinflammatory genes (Figure 1), macrophages in the periphery and microglia in the CNS release cytokines and chemokines as a function of the innate immune system [99]. Cytokines are small signaling molecules that control the activation of other immune cells, and can be proinflammatory (IL-1, IL-6, TNFα, IL17, IFN, etc.) or anti-inflammatory (IL12 and IL6) [100]. Interferons (IFNs) are antiviral cytokines that can be activated by any pathogen. IFNs bind to immune cells and activate transcription of antiviral genes [101]. These include antiviral effectors as well as positive and negative regulators of the innate immune response [102]. Chemokines are small signaling molecules that bind to G-protein-coupled receptors on immune cells to mobilize them to areas of injury. Cytokines produced by the dam can cross through the placenta and blood–brain barrier to the fetal brain [103,104], or can cause injury to the placenta, which can alter blood flow to the fetus, causing tissue hypoxia and exacerbating injury (Figure 2) [39]. In particular, the proinflammatory cytokine IL-6 has been implicated in the development of abnormal behavioral changes, as a study in rats demonstrated that a single injection of IL-6 at gestational day 12.5 led to sensorimotor processing deficits in offspring that is often associated with a schizophrenic phenotype [103]. IL-6 may preferentially transfer from the mother to the fetus at specific periods of gestation. Radiolabeled IL-6 was able to cross into rat pups at mid gestation; however, it was not able to cross the placenta when injected in late gestation [105]. Furthermore, mouse pups exposed to Poly(I:C) at E12.5 had increased cytokine mRNA in the placenta, as well as increased cytokine expression in the brain [106]. Maternal cytokines can trigger maternal Th17 cells which release IL-17a, which has been shown to result in a cortical disorganization when injected at gestational day 18.5 in mice [107]. In the same study, MIA and IL-17a were associated with abnormal levels of proteins associated with cortical organization such as SATB2, TBR1, and CTIP2.

Newer research has found a more direct effect of MIA on the developing fetus. It was recently found that LPS can cross the placental barrier to enter the fetal compartment [108]. In CD-1 mice, FITC-labeled LPS was found in the maternal uterus, placenta, and fetal tissue after intravenous administration. This result was also replicated using ^125^I-labeled LPS: when injected into pregnant mice, ^125^I-LPS was found within the placental and the fetal tissues [109]. This indicates that direct LPS, or perhaps other pathogens within the fetus, may also contribute to the neurodevelopmental effects of early neuroinflammation. Following either transmission of cytokines to the fetal brain or direct LPS transmission, fetal microglia enter a proinflammatory state, releasing additional cytokines and chemokines [110]. The exact mechanisms by which maternal immune activation impacts the pathology of HIE and other long-term health consequences require further study [40,111,112].

Gestational age impacts the response to MIA [52,113]. Maternal infection has also been shown to increase the risk of premature birth, which has large cumulative impacts on neonatal health [113]. Infants exposed to MIA at a younger gestational age have poorer neurological outcomes when compared to infants exposed closer to term, though all infants exposed to MIA had worse neurological outcomes compared to infants who were not exposed [52]. There are also differences in the immune response in the preterm infant and the term infant. The fetal immune system is immature, as the adaptive immune system (responsible for specific pathogen responses and memory) does not develop until after birth [114]. As such, infants earlier in gestation are more susceptible to infection and its negative outcomes, as demonstrated by the increased risk of sepsis, with 60% of infants born at less than 25 weeks EGA (estimated gestational age) developing sepsis compared to 33% of infants born at 28 weeks EGA [115]. Preterm infants also have less complement activity in the serum compared to term infants [116,117]. Cumulatively, this results in significant impairment in the preterm infants’ ability to phagocytose pathogens and generate an inflammatory response, as well as a reduced ability to recruit other immune cells such as leukocytes and lymphocytes to the site of infection. Additionally, neutrophils in the preterm infant migrate more slowly and in decreased numbers compared to term infants, whose neutrophil migration does not differ from adults [118]. However, in a study of preterm infants exposed to infection, IL-6 was significantly elevated, suggesting that preterm infants are able to respond robustly to inflammatory conditions [119]. Overall, as gestational age decreases, the ability to mount a successful immune defense against pathogens decreases due to the immaturity of the immune system response, including the complement system, neutrophil migration, and cytokine response [117].

Activation of fetal microglia, the resident macrophages of the brain, by either cytokine transmission or direct pathogen binding are an important factor in neurodevelopmental disruption following MIA. In the embryonic and postnatal period, microglia exhibit a more ameboid phenotype, characterized by a larger soma and smaller projections, compared to the ramified phenotype in mature microglia, characterized by smaller soma size and longer, more complex projections [120]. Microglia play a critical role in neurodevelopmental processes such as synaptic pruning and neural precursor cell regulation (Figure 3) [121,122]. Studies have demonstrated that there are several mechanisms through which microglia regulate synapse formation [123,124]. One such mechanism is phagocytosis mediated by the fractalkine receptor CX3CR1 [125]. More recently, microglia triggering receptor expressed on myeloid cells 2 (TREM2) has been implicated in functional brain connectivity, with mice lacking this receptor experiencing pathogenic excitatory transmission and deficits in social behavior [123].

Following MIA, fetal microglia exhibit an increase in the ameboid phenotype typical of either activated or immature microglia (Figure 3) [110]. A single-cell RNAseq study of fetal brains following MIA found that microglia exhibit transcriptional changes that are associated with microglial activation, including an upregulation of Ccl12, Jund, Isg15, and Clec12a [126]. These microglial changes are associated with the release of additional cytokines and immune factors, thereby exacerbating the inflammatory response and its deleterious effects [110,127]. The activation of microglia during important periods of neurodevelopment also has direct impacts within the brain. There are many neurodevelopmental disruptions primarily within the cortex following MIA. These include over-proliferation of excitatory cortical neurons, decreased neurogenesis of inhibitory neurons, hypomyelination, and disrupted synaptic development and function [110,128,129,130]. These neurodevelopmental changes may be a consequence of microglia activation as they are highly involved in these processes [110,130]. Studies investigating the underlying causes of increased susceptibility to seizures following MIA have found many functional changes, including reactive astrogliosis, increases in hippocampal baseline excitability, shifts in EEG spectral power, imbalanced expression of ion transporters, and persistent depolarizing action of GABA [84,86,87,88,131]. These functional differences are likely due to these microglia-mediated neurodevelopmental alterations. The direct role of microglia is supported by the rescue of both behavioral and synaptic alterations following MIA by the depletion and repopulation of microglia [132]. Microglia also regulate the response of other cortical cells to MIA [126].

**Figure 3 biomedicines-12-02559-f003:**
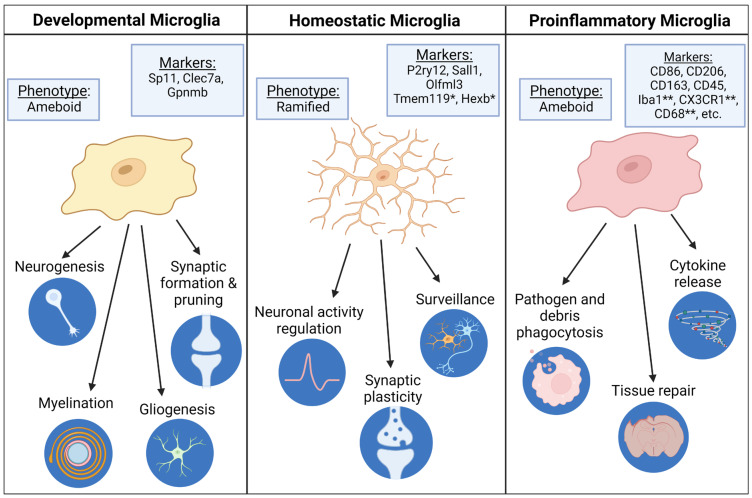
Microglia exhibit different phenotypes, transcriptional markers, and functions depending on developmental timing and activation state. There are some caveats to the transcriptional markers presented in Figure 3. Most developmental microglia have transcriptomes similar to homeostatic microglia. However, a subset referred to as proliferative-region-associated microglia (PAMs) have distinct transcriptional markers referenced here [133]. * Tmem119 and Hexb are often referred to as homeostatic markers. However, the expression of these genes does not change in proinflammatory microglia [134]. Therefore, it is more accurate to refer to these as general microglia markers. ** Many commonly used microglia identifiers are upregulated in proinflammatory microglia [135].

Microglia alterations are not the only possible mechanism of neurodevelopmental disruption following MIA (Figure 2) [136]. Inflammation has been shown to cause oxidative stress via reactive oxygen species (ROS) imbalances in the brain [137] and has been implicated with the development of ASD and schizophrenia [138,139]. This oxidative stress, in combination with other factors, can affect neurodevelopment by epigenetic changes, excitotoxicity, apoptosis, and mitochondrial dysfunction [136,139,140]. After an injection of LPS to a pregnant rat, pups were shown to have increased markers of oxidative stress in both the brain and the placenta, based on elevated malondialdehyde and carbonyl protein formation [104]. This demonstrates that maternal immune activation alone is enough to trigger an oxidative stress response [136]. Mitochondrial dysfunction in particular has been linked with ASD, with studies showing that people with ASD may already be more susceptible to oxidative stress as compared to healthy controls [141]. Mitochondrial dysfunction that can arise from oxidative stress can be due to DNA damage and altered proteins [142]. Further, dysfunctional mitochondria can further exacerbate already-existing oxidative stress, as mitochondria are a major source of ROS via the electron transport chain [143], creating a maladaptive feedback loop. Microglia have been implicated in this process, as activated microglia have been shown to increase ROS in the brain [144] and to play a role in NOX signaling, which is critical for regulating the oxidative stress response [145].

Although maternal infection is a known risk factor for developmental delays and disorders in humans, most cases of maternal infection or inflammation do not lead to these outcomes. Therefore, it is hypothesized that maternal immune activation may have a priming effect which leads to increased susceptibility to genetic or environmental “second hits”, further increasing the risk of developing various CNS disorders [79,146,147,148,149] depending on the timing, severity, and type of cumulative stressors [150]. In a study using a dual-hit model in mice, pregnant dams were given Poly(I:C) or saline. On postnatal day thirty or forty, offspring were exposed to unpredictable stress [151]. In adulthood, neither MIA nor unpredictable stress alone were enough to disrupt sensorimotor gating or increase their sensitivity to psychomimetic drugs; however, animals exposed to the dual hit demonstrated disruptions in both. In a two-hit model of prenatal and postnatal inflammation, only the presence of both hits was sufficient to induce spontaneous bursts of epileptiform activity [85]. The hypothesis that maternal immune activation can interact with other insults and immune challenges to increase deleterious outcomes helps to explain the significance of maternal inflammation in the rates of HIE.

The compounding effects of perinatal inflammation and later insults may be largely due to disruptions in microglial function. Microglia exhibit a blunted response to later immune challenges following MIA [110,152]. When microglia do not properly mount their proinflammatory response, pathogens and cellular debris are not cleared properly [153]. Prior inflammation can also be a driver in neurodevelopmental pathology through the infiltration of peripheral macrophages and other immune cells following the initial inflammatory hit. Studies have shown that the activation of TLR3 and TLR4 can have a cascade of consequences on the brain prior to injury, one of them being the weakening of the blood–brain barrier that allows peripheral immune cells to infiltrate the brain parenchyma [154]. These infiltrating peripheral macrophages (CCR2+) preferentially adopt a proinflammatory (CD68+, TNFα+) phenotype [155]. Over the course of several months, these cells can adopt a microglia-like phenotype (Tmem119+, Sall1+, or P2RY12+) or maintain their TNFα+MHCII+ phenotype.

Sex differences within MIA outcomes have also been noted, particularly in rodent models, with males often experiencing worse outcomes [156,157]. Males have been shown to exhibit more severe long-term behavioral changes than females, including increases in anxiety-like and depressive-like behaviors [158], as well as ASD-related behaviors such as social deficits [159,160]. Males exhibit larger deleterious effects in the placenta such as increased hemorrhage and growth restriction [161]. The hypoxic effects of MIA within the fetal cortex have also been shown to be worse in males [161]. There have been several proposed mechanisms for male susceptibility or female protection following MIA. Differing immune responses to MIA within male and female animals are likely involved. Males exhibit larger short-term cytokine responses to maternal immune activation in the placenta and the fetal brain [106,161]. This increased immune response may be due to the increased activation of microglia observed in males [162]. When cytokines were analyzed at a later timepoint, males expressed lower cytokine levels compared to females, suggesting that their baseline immunity may be decreased [158]. This is supported by a finding in adult mice in which males had decreased expression of CD68, a marker of phagocytic microglia [163]. Males also exhibited more significant changes in genes related to synaptic plasticity in the cortex after MIA [158]. The estrogen receptor ERβ has been proposed as a potential mediator of the sex differences in the fetal inflammatory response [164]. ERβ regulates the inflammatory response of microglia, as demonstrated through exaggerated inflammatory responses following reduced ERβ expression [165]. There may be sex-dependent patterns of ERβ expression in fetal brains, with decreased ERβ expression in female amygdala embryonic neurons [164]. This may also not be due to ERβ expression itself, but the sexually dependent expression of, and response to, steroid ligands. Additionally, there may be sex differences in the response to invading peripheral immune mediators such as regulatory T-cells following immune activation or injury. In females, invading regulatory T-cells have been shown to be neuroprotective, while in males, they promote neurodegeneration [166]. However, it should be noted that not all studies have found sex differences in behavioral outcomes or immune response within MIA models [167].

## 5. Commonly Used Animal Models of HIE and the Incorporation of MIA and Inflammation

Large animal models have proven critical to the field of neonatal brain research and have their own benefits that deepen clinical understanding of HIE. Sheep models of HIE are achieved via umbilical cord occlusion [168,169] or maternal hypoxia [155,170]. Umbilical cord occlusion typically occurs for at least 10 min, though there is variability in protocols between research groups [168,169,171]. Fetal oxygen levels can be titrated via modulating maternal oxygen levels and then measuring fetal blood flow and oxygen levels. These experiments are typically conducted on preterm sheep, at around 0.8–0.9 of gestation, due to having a brain growth trajectory that is more advanced than human infants [172,173]. To model HIE for preterm infants, 0.65-term sheep can be used [172,173]. As fetal sheep remain in the uterine environment during injury, this model optimally recapitulates the maternal–placental–fetal interface. Several studies within large animal models have utilized LPS to recapitulate the inflammatory environment of HIE [174,175]. LPS given to pregnant ewes via bolus injection was shown to decrease fetal O_2_, increase fetal PaCO_2_, and decrease fetal pH, reflecting a hypoxemic state with metabolic acidosis [176]. Chronic exposure to an inflammatory environment has been shown to alter white matter development within fetal sheep [175]. Studies such as this could give insight as to the mechanisms through which maternal infection could lead to a hypoxic–ischemic event through close monitoring of the fetus without losing the maternal interface. However, there is the drawback that these animal models focus solely on that of maternal immune activation without the presence of an acute hypoxic event.

Pigs have also been used to model HIE and are of particular interest due to both their white matter development and brain growth development, which closely mimic that of human infants [172]. One pig HIE model is achieved using anesthetized newborn piglets mechanically ventilated with 6% oxygen for 17–55 min, until they reach an EEG amplitude below 7 microvolts (a measure of encephalopathy) [177]. Another model used in pigs is achieved via occlusion of bilateral carotid arteries and subsequent hypoxia. This procedure utilizes phosphorus magnetic resonance spectroscopy in order to achieve low cerebral ATP levels [112]. Another piglet model uses hypoxia followed by complete asphyxia, following the protocol of 10% O_2_ for 30–45 min followed by 7–8 min of airway occlusion [172]. The intense monitoring of the piglets required by these models is an advantage as the hypoxic event can be highly titrated and consistent; however, the piglet models are all achieved extrauterine, and therefore, the maternal and placental environment cannot be replicated or investigated fully (Figure 4). The incorporation of prenatal inflammation and infection into pig models of HIE is a newer development, with a two-hit model being developed in 2019. In this model, LPS is infused into newborn pigs starting 4 h prior to hypoxia, with continued infusions for 48 h after the hypoxic event [178]. Newborn pigs that were primed with LPS showed greater microglial proliferation and astrogliosis compared to pigs who underwent hypoxia alone. This finding is particularly salient in regard to treatment of HIE, as the same model demonstrated that within LPS sensitized newborn piglets, therapeutic hypothermia was less effective as a treatment compared to those who were not sensitized [179]. Since therapeutic hypothermia is the only effective treatment for HIE, environmental and clinical factors that affect its efficacy are critical to examine further. A limitation of this study and model is that while these newborn pigs were exposed to infection, the maternal–fetal interface was lost as the transfusions occurred in the absence of the maternal compartment.

Primates have also been used to model HIE and have an advantage of most closely approximating human brain development (Figure 4). The primary model used today is the umbilical cord occlusion model, in which fetuses are surgically born and then subjected to umbilical cord occlusion lasting between 15 and 18 min [180]. Other primate models employ the use of saline-filled rubber sacs placed over the head of the fetus to induce total asphyxia in addition to umbilical cord occlusion or high-dose-oxytocin-induced prolonged partial asphyxia [181]. The major benefit to nonhuman primate models is their similar brain development and complexity compared to other large animals or rodents, with injured nonhuman primates demonstrating similar white matter injuries to that of human infants based on both MRI and histopathological results [182]. Nonhuman primates also show an increase in activated microglia and gliosis after brain injury [183]. These similarities between human infants and nonhuman primates could lead to more promising treatments, such as determining potential biomarkers of injury severity [184]. The effect of MIA on neurodevelopment has been explored in primate models [185,186,187,188], with the primary model being an intravenous Poly:IC injection into pregnant rhesus macaques [188]. Primates exposed to MIA demonstrated repetitive social behaviors similar to those seen in rodents [188], as well as elevated cytokines that were associated with these behaviors and persist past the neonatal period [185]. However, there is a lack of studies and primate models that incorporate MIA in conjunction with an acute hypoxic–ischemic injury, and this is a limitation of current primate models of HIE.

There are drawbacks to using larger animals and non-human primates as a model of HIE, particularly due to the difficulty of maintenance of the animals, which are costly to house (Figure 4). The techniques involved in these models are also technically challenging, sometimes requiring around-the-clock ICU care, and many institutions do not have the resources required to perform them [189]. There are also ethical considerations for large animal models, particularly for the nonhuman primate models. Furthermore, molecular and cellular tools have not been developed for larger animals as they have been for rodent models. This limits the ability to investigate the downstream molecular and cellular mechanisms of HIE within the larger animal model systems.

Rodents and small animals also provide valuable insight into the mechanisms of HIE. The current standard rodent model of HIE is the Rice–Vannucci model, in which a postnatal mouse or rat is subjected to a unilateral carotid artery ligation followed by global hypoxia (typically 8% oxygen) for 90 min [190]. The timing of this injury varies; however, it is often performed between postnatal day 7 and 10, which is considered equivalent to full-term human gestation brain development [191]. The result of this insult is a large unilateral hemispheric injury to the ipsilateral hemisphere with a significant amount of cell death. Mice subjected to this model have an injury that mimics a large unilateral cerebral infarction, with the expected hemiplegic motor deficits [192,193]. While Rice–Vannucci is the standard model, its clinical relevance has been questioned due to the nature of the induced injury. MRI scans of human neonates with HIE often show more global white matter and deep gray matter (basal ganglia, thalamus, and pons) injury unlike that of a hemispheric cerebral infarction seen in rodent models [194,195]. Mild HIE has also been associated with injury to watershed regions, not produced within the Rice–Vannucci model [196]. It is also possible for human neonates to have several injury patterns or diffuse cortical hypoxia. Due to these significant differences in injury patterns, the need to create rodent models beyond the Rice–Vannucci model is significant. It is also important to note that this model is achieved entirely using postnatal animals; therefore, it cannot replicate any maternal or placental factors that impact HIE and its outcome, and thus it could be missing key aspects of injury pathophysiology. The Rice–Vannucci model is not the only used model of HIE in rodents. Some models have found motor outcomes similar to cerebral palsy utilizing acute hypoxia [197,198]. Occlusion of the middle cerebral artery in neonatal mice and rats has also been used as an alternative to carotid artery ligation [192,199]. Some rodent models have used intrauterine hypoxia–ischemia, although not always in the context of HIE [200,201]. These limited animal models often produce more mild-to-moderate results in comparison to the more severe results of the Rice–Vannucci model.

Rodents are not the only small animal model that has been used to investigate HIE. In recent years, the use of ferrets as a model of HIE and infection has been utilized, as ferrets have the advantage of a similar white-to-gray-matter ratio and gyrencephalic cerebral cortex to infants that is not seen in rodent models [202,203]. Ferrets also pose less of a burden to maintain than larger animals such as pigs and sheep. The current model is achieved by injecting P10 ferret pups with LPS, followed by hypoxia and then hyperoxia. Ferret pups who underwent this procedure demonstrate developmental delays, abnormal gait, and white matter injury [204]. It is important to note that P10 ferret pups are considered equivalent to 24 weeks gestation in humans [202]. Brain injury outcomes vary based on gestational age, and preterm HIE is associated with cognitive and sensory deficits and white matter injury rather than the motor deficits and watershed injury commonly seen in term gestation HIE [205,206]. This model is also at a disadvantage in that it fails to replicate the maternal–fetal–placental interface, as both the injury and the immune activation are performed outside of the maternal environment.

Several new models of HIE within rodents incorporate the intrauterine inflammatory environment using MIA. Most of these models utilized systemic immune activation via an injection of LPS or Poly(I:C) into the dam, followed by a hypoxic–ischemic event, such as the Rice–Vannucci procedure. In rats, pups sensitized with a maternal LPS injection had a more robust microglia-mediated inflammatory response compared to pups that were not MIA-sensitized [207]. This indicates that MIA may sensitize microglia and lead to poorer neurological outcomes [207]. Other studies found that therapeutic hypothermia proved ineffective in pups immune-sensitized by direct LPS administration hours prior to Rice–Vannucci administration [208,209]. Environmental factors that impact the benefits of therapeutic hypothermia, such as intrauterine infection, should be focused areas of intensive study. Our laboratory recently developed a novel mouse model of neonatal HIE that both captures the inflammatory intrauterine environment commonly seen in human neonates as well as more accurately replicates the injury itself [2,210]. This is achieved via a two-hit model of maternal immune activation at late gestation followed by postnatal global hypoxia at term equivalence. This model allows investigation of both the maternal–placental–fetal system of inflammation, followed by a more clinically relevant global hypoxia that is akin to the pathophysiology in human neonates with HIE. These two hits result in developmental delays, motor impairments in adulthood, neuroinflammation, and transcriptional changes in microglia.

## 6. Considerations for the Incorporation of MIA into Animal Models of HIE

Investigators have begun to incorporate MIA or a direct immune priming of neonates into animal models of HIE. There are many variables that should be considered when creating these animal models. LPS has largely been used in HIE animal models due to the importance of chorioamnionitis as a risk factor, whereas the viral mimetic poly(I:C) may be less relevant for HIE [1]. Direct immune activation of pups has been used in animals as an immune priming prior to HIE insult. However, this loses maternal and placental interaction, and the use of MIA more translationally represents risk factors of HIE.

The dosage and timing of the injection of immune modulators is an important consideration, as both variables range significantly in the literature. In rodents LPS doses range from 25 μg/kg to over 1 mg/kg, although 100 μg/kg is the most frequently used dose [211,212]. Poly(I:C) doses in rodents typically range from 4 mg/kg to 20 mg/kg [211,212]. There are significantly fewer studies on the use of maternal immune activation in other animal models. In MIA models within pigs, poly(I:C) has been used at a dose of 1.0 mg/kg or 0.5 mg/kg [213]. PRRSV, a swine-specific virus, has also been used at 5 mL 1 × 10^5^ 50% tissue culture infectious dose (TCID50) [214,215]. LPS has been used at a dose of 2 μg/kg in pregnant gilts, although not within the context of MIA [216,217]. Further research would need to be conducted to determine if this dosage would be sufficient to elicit changes in offspring. One experiment in sheep utilized systemic LPS via the maternal JV catheter at 2 μg/kg of maternal body weight [176].

The injection of LPS into pregnant animals can cause complications that should be weighed. LPS is abortifacient in rodents [218,219]. In rodents, this may result in fetuses being reabsorbed prior to birth, particularly if the MIA is earlier in gestation; or it may result in non-viable pups at birth, particularly in the case of late gestation MIA [218]. Our experience with LPS at late gestation in mice revealed higher maternal hemorrhage and greater fetal loss at higher LPS dosages (>100 µg/kg), ultimately leading us to choose very lose dose LPS (50 µg/kg) for E18 MIA exposure [210]. The abortifacient impact of LPS has also been recorded in pigs, and so should be taken into consideration in large animal models [220]. Higher doses of LPS will cause higher abortifacient outcomes. Therefore, the dosage of LPS should be trialed to minimize fetal reabsorption and death, while maintaining cellular and molecular changes and behavioral phenotype.

The timing of both poly(I:C) and LPS can range all through gestation in rodents. However, poly(I:C) is most used in mid-gestation (E12.5 and E15), while LPS is most used either in mid (E15) or late (E18) gestation [218,219]. This reflects the differential susceptibility of the fetus to viral vs. bacterial infection at different gestational stages. Some viral infections such as zika virus and cytomegalovirus may be more deleterious in earlier gestation due to the effects on neural progenitor cells and cortical migration [221,222,223]. However, bacterial infections in the second and third trimester are particularly linked to ASD diagnosis [64]. Late gestational infection may be relevant to the timing of infection as a risk factor for HIE. Animals all have different rates of nervous system development relative to humans [224]. The specific timing of neural development such as gliogenesis, neurogenesis, myelination, synaptogenesis, synaptic pruning, and apoptosis should all be taken into consideration in the development of these models.

The methodology of hypoxic-ischemic insult should also be considered. Larger animal models more accurately replicate the human injury through umbilical cord occlusion, maternal hypoxia, or asphyxiation [172]. However, size or physiology can make these methods unusable or less effective in rodents. The Rice-Vannucci rodent model causes significant motor and cognitive impairments, but the hemispheric injury produced by the unilateral carotid artery ligation is not the best reflection of injury pattern in humans [193,194]. Some more recent rodent models have stepped away from this unilateral ligation through various techniques including maternal hypoxia, pup hypoxia in isolation, and cardiac arrest [192,197,225]. However, the phenotypes produced by these models are often less severe than the Rice-Vannucci model. While they might not be the best reflection of symptoms such as severe cerebral palsy, these models may more accurately represent mild to moderate HIE. The inclusion of risk factors such as MIA may serve to increase the severity of these models. HIE is an incredibly heterogenous injury, and investigators should weigh the type and severity of injury that will allow for them to answer the specific questions they seek to investigate.

## 7. Conclusions

The outcomes of HIE vary significantly depending on the risk factors present within the maternal–fetal interface, the type and extent of the hypoxic–ischemic event, and the resulting pattern of injury. Therapeutic hypothermia is not always effective in counteracting the devastating effects of HIE, either due to the limited timing of therapeutic efficacy or to individual variability. While this treatment was discovered in large animal models, the animal size and limited cellular and molecular tools pose many limitations and make these models inaccessible to many labs. Rodent models have greatly contributed to understanding the pathophysiology of HIE. However, none of the treatments that were promising in rodent models have proved effective in humans. This points to a significant need to develop animal models that recapitulate the many nuanced factors that contribute to the pathophysiology of disease and efficacy of treatment, while being accessible. MIA is an environmental factor that significantly contributes to the development and outcomes of HIE. MIA is easy to incorporate into HIE models, as it is already a commonly used rodent model of neurodevelopmental disorders such as schizophrenia and ASD, as well as a contributing factor to epilepsy susceptibility. MIA is non-invasive and involves a peripheral injection of non-infectious proinflammatory molecules, namely LPS or Poly(I:C). The mechanisms and outcomes of MIA have also been well characterized. The cognitive and behavioral deficits observed in models of HIE are primarily due to the activation of microglia and a disruption of their role within neurogenesis and synaptic development. MIA has also been shown to compound with second hits to create worse outcomes than either insult in isolation. The incorporation of MIA or inflammation in animal models of HIE has shown reduced efficacy of therapeutic hypothermia, possibly explaining some of the variability in humans. The cellular and molecular mechanisms involved in the interaction of MIA and HIE should be further investigated, and these models can be used as a more accurate way to investigate new therapies. The preclinical data obtained from the development of new animal models will hopefully lead to more personalized treatments and better outcomes for the children affected by this common and devastating disease.

## Figures and Tables

**Figure 1 biomedicines-12-02559-f001:**
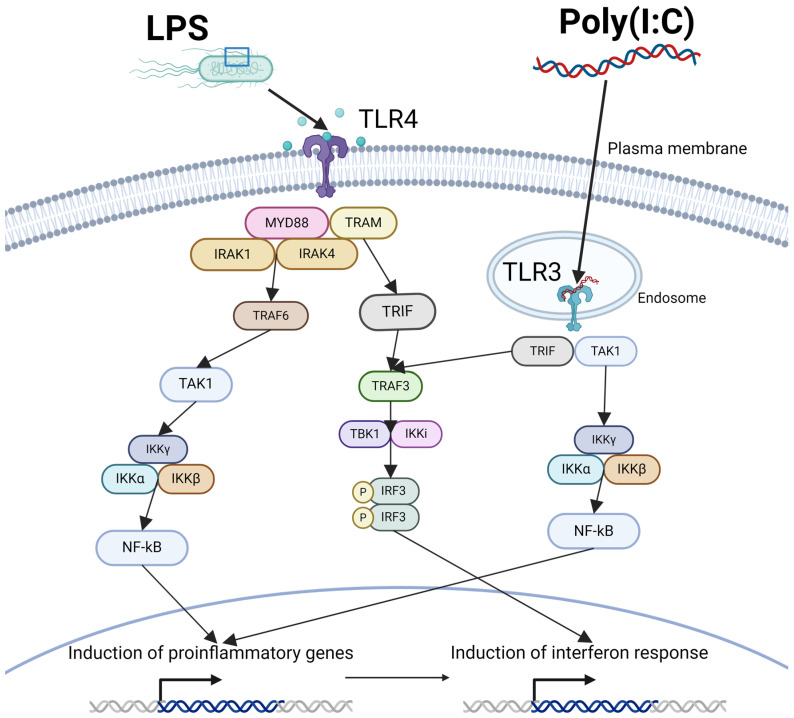
LPS and Poly(I:C) bind to toll-like receptors, which initiates intracellular pathways, initiating the transcription of proinflammatory genes.

**Figure 2 biomedicines-12-02559-f002:**
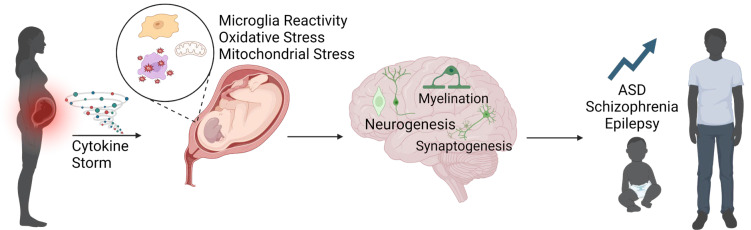
MIA initiates inflammation, which disrupts neurodevelopmental processes, leading to increased rates of ASD, schizophrenia, and epilepsy.

**Figure 4 biomedicines-12-02559-f004:**
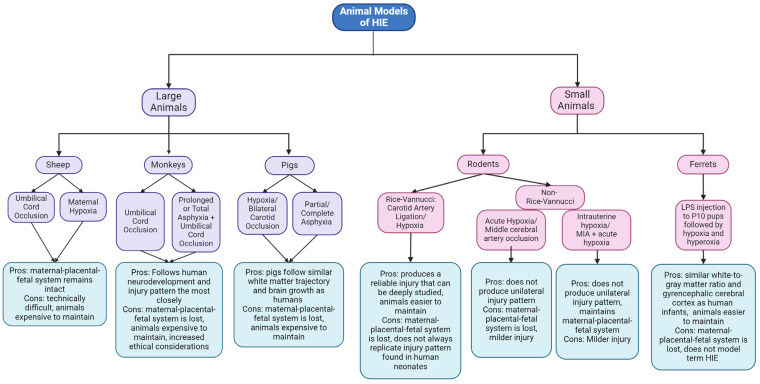
HIE model comparison.

## Data Availability

Not applicable.

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
