# Peer review of "The Importance of Including Maternal Immune Activation in Animal Models of Hypoxic–Ischemic Encephalopathy"

_biomedicines, 2024, doi:10.3390/biomedicines12112559_

Round 1

Reviewer 1 Report

Comments and Suggestions for Authors

REVIEW

In full-term newborns, an important cause of impaired neurological development is hypoxic-ischemic encephalopathy (HIE). Early neonatal brain damage is an important cause of mortality and morbidity, often resulting in lifelong consequences. The mechanisms associated with the development, course and regeneration after neonatal brain damage and their impact on the disruption of further development of the immature brain have not yet been satisfactorily explained.

Title

The title of the work reflects the scope of the research conducted.

Abstract

The abstract requires correction and supplemented with a final conclusion.

Introduction

The introduction to the research topic is too short and should be expanded. I believe that focusing on the results of studies conducted on rodents is burdened with great uncertainty due to the different physiology of the gastrointestinal tract, reproductive system and secretory system. I believe that using the pig as an animal model in the study of human diseases, including hypoxic-ischemic encephalopathy (HIE), is the most reliable model in the study of HIE. Due to the striking similarities in organ systems and high availability, pig organs also best meet the criteria for xenotransplantation. For many years, pig tissues have been used, among others, in the treatment of heart valve defects in humans. However, pigs offer much more than just a potential source of organs. Due to the high similarity in the anatomy of the gastrointestinal tract and the physiology of digestion, pigs are also the best candidate for research on human nutrition and the pharmacokinetics of orally administered drugs. In addition, the prenatal and postnatal development of the digestive and nervous systems is very similar in both species, which is why newborn piglets are often used as an animal model for human newborns. For example, in our own experiments on prematurely born piglets, the influence of bioactive components of mother's milk on brain growth and development was studied. Similarities in the structure and function of the pancreas itself make the pig also an attractive model for research on exocrine pancreatic insufficiency (EPI or NTZ) and type 1 or 2 diabetes. Humans and pigs also share many features in the structure and function of the kidneys and the lower urinary tract, which was used, among others, in our own research on the development of enzymatic therapies in the treatment of kidney stones and hyperuricemia, i.e. elevated levels of uric acid in the blood. The research topic undertaken by the authors is of great scientific value in order to understand the mechanisms of the development of hypoxic-ischemic encephalopathy (HIE), diagnosis and course of this disease. The introduction should clearly state the purpose of this article.

Discussion

This chapter is well written but should be supplemented with information provided in the review summary and a request to systematize it.

Review summary

A number of studies conducted over the past few years have shown that not only the correct experimental protocol, but also the selection of the appropriate animal model can determine its result and the correct interpretation of the data obtained. Due to the rapidly increasing average life expectancy of humans, and consequently the more frequent occurrence of chronic diseases, neurodegenerative diseases, the importance of translational research on animal models in modern medicine is increasing. The article is interesting but should be rewritten and supplemented. It would be worth mentioning in the article modern non-invasive techniques analyzing the proteome and metabolome, which can contribute to understanding the predispositions to the development, course of better diagnosis and treatment of brain disorders. The use of advanced magnetic resonance imaging (MRI) techniques: DWI, ASL, allows for visualization of neuronal density, synaptogenesis and deviations in myelination. Proton magnetic resonance spectroscopy (1H-MRS) enables noninvasive measurement of neurometabolite concentrations in vivo. Additionally, these nonconventional MRI techniques enable quantitative assessment and can provide numerical data on neurometabolite concentration, neuronal density, and myelination level, thus providing a basis for building multidimensional models that take into account clinical, omics, and radiological data. The authors concluded that there is a significant need to develop animal models that reflect multiple factors contributing to disease pathophysiology and treatment efficacy.

The manuscript of the article contains the names of three authors and the abstract on the MDPI submission page contains 2 names. Please explain.

Author Response

Reviewer 1:

REVIEW

In full-term newborns, an important cause of impaired neurological development is hypoxic-ischemic encephalopathy (HIE). Early neonatal brain damage is an important cause of mortality and morbidity, often resulting in lifelong consequences. The mechanisms associated with the development, course and regeneration after neonatal brain damage and their impact on the disruption of further development of the immature brain have not yet been satisfactorily explained.

Title

The title of the work reflects the scope of the research conducted.

Abstract

The abstract requires correction and supplemented with a final conclusion.

  • Thank you for this comment. We understand that the focus and conclusions of this paper were perhaps not made clear within the abstract and introduction. Our intent with this review is to highlight the importance of risk factors for HIE in the creation of translational animal models, while specifically discussing the relevance of maternal immune activation (MIA). There is a large amount of research about MIA, primarily in rodent models, that researchers focused on HIE may not be aware of, and so we summarized this literature. To clarify this focus, we have expanded our abstract (lines 25-31).

Introduction

The introduction to the research topic is too short and should be expanded. I believe that focusing on the results of studies conducted on rodents is burdened with great uncertainty due to the different physiology of the gastrointestinal tract, reproductive system and secretory system. I believe that using the pig as an animal model in the study of human diseases, including hypoxic-ischemic encephalopathy (HIE), is the most reliable model in the study of HIE. Due to the striking similarities in organ systems and high availability, pig organs also best meet the criteria for xenotransplantation. For many years, pig tissues have been used, among others, in the treatment of heart valve defects in humans. However, pigs offer much more than just a potential source of organs. Due to the high similarity in the anatomy of the gastrointestinal tract and the physiology of digestion, pigs are also the best candidate for research on human nutrition and the pharmacokinetics of orally administered drugs. In addition, the prenatal and postnatal development of the digestive and nervous systems is very similar in both species, which is why newborn piglets are often used as an animal model for human newborns. For example, in our own experiments on prematurely born piglets, the influence of bioactive components of mother's milk on brain growth and development was studied. Similarities in the structure and function of the pancreas itself make the pig also an attractive model for research on exocrine pancreatic insufficiency (EPI or NTZ) and type 1 or 2 diabetes. Humans and pigs also share many features in the structure and function of the kidneys and the lower urinary tract, which was used, among others, in our own research on the development of enzymatic therapies in the treatment of kidney stones and hyperuricemia, i.e. elevated levels of uric acid in the blood. The research topic undertaken by the authors is of great scientific value in order to understand the mechanisms of the development of hypoxic-ischemic encephalopathy (HIE), diagnosis and course of this disease. The introduction should clearly state the purpose of this article.

  • We have expanded the introduction to the research topic and have included a summary of the topics and conclusions presented in this paper (lines 51-122). Thank you for this suggestion. Our focus on rodents is quite large within the discussion of maternal immune activation (MIA) research. This is due to the majority of the research in this area taking place in rodents. In the discussion of HIE, we have weighed the positives and negatives of each type of animal model. We have also added a section at the end of this paper advising researchers who may want to incorporate MIA into their HIE models on what variables should be taken into consideration (section 6, lines 538-595). We agree that large animal models such as pigs are likely much closer to humans in regard to brain development and organ function. However, the large burden of resources, cost, and care of these animals make them inaccessible to many labs. Some labs may additionally choose rodent models due to the many molecular and transgenic possibilities available, specifically for mice. We believe that the use of many different animal models can contribute meaningfully to our understanding of HIE and the efficacy of treatments. Thank you for appreciating the scientific value of this review.

Discussion

This chapter is well written but should be supplemented with information provided in the review summary and a request to systematize it.

  • What were previously sections 2 (Benefits and limitations of established HIE animal models) and sections 6 (Animal models of HIA can incorporate MIA or inflammation), have been combined into section 5 (Commonly used animal models of HIE and the incorporation of MIA and inflammation). Section 5 now is organized by animal model and discusses the typically used methodology for HIE models followed by how some studies have incorporated MIA or inflammatory priming. This allows us to discuss HIE animal models and HIE models that include inflammatory priming in one section, as well as clarify the focus of the review on maternal immune activation. On lines 496-502 we also added some non-Rice-Vannucci rodent models of HIE and have added those to Figure 4 as well. We have addressed the comments of the review summary below. However, if there are specific organizational changes that you think would benefit the clarity of the paper, please let us know.

Review summary

A number of studies conducted over the past few years have shown that not only the correct experimental protocol, but also the selection of the appropriate animal model can determine its result and the correct interpretation of the data obtained. Due to the rapidly increasing average life expectancy of humans, and consequently the more frequent occurrence of chronic diseases, neurodegenerative diseases, the importance of translational research on animal models in modern medicine is increasing. The article is interesting but should be rewritten and supplemented. It would be worth mentioning in the article modern non-invasive techniques analyzing the proteome and metabolome, which can contribute to understanding the predispositions to the development, course of better diagnosis and treatment of brain disorders. The use of advanced magnetic resonance imaging (MRI) techniques: DWI, ASL, allows for visualization of neuronal density, synaptogenesis and deviations in myelination. Proton magnetic resonance spectroscopy (1H-MRS) enables noninvasive measurement of neurometabolite concentrations in vivo. Additionally, these nonconventional MRI techniques enable quantitative assessment and can provide numerical data on neurometabolite concentration, neuronal density, and myelination level, thus providing a basis for building multidimensional models that take into account clinical, omics, and radiological data. The authors concluded that there is a significant need to develop animal models that reflect multiple factors contributing to disease pathophysiology and treatment efficacy.

  • We absolutely agree that experimental protocol, animal model selection, and type of data collected can all impact results and data interpretation. Translatability should be a high priority in the use of animal models to investigate diseases. While we appreciate the discussion of the modern non-invasive techniques you mention here, we feel that this is outside the scope of our current review. Our review is focused on the importance of maternal immune activation in HIE and neurodevelopmental disorders, and how it can and has been incorporated in animal models of HIE. Incorporating techniques to investigate HIE models in general would add significantly to the length and scope of this paper, and we feel that it would dilute the central message.

The manuscript of the article contains the names of three authors and the abstract on the MDPI submission page contains 2 names. Please explain.

  • A third author was added between initial submission and review. An author change form was submitted. However, the submission page may not have been updated, leading to this discrepancy.

Reviewer 2 Report

Comments and Suggestions for Authors

1.      Authors are advised to check their references regarding the inflammatory markers.

2.      Do authors propose that the inflammatory reaction/mechanisms among the various animal models are similar. Does the distinction among the models made in Fig.1 has no impact? Please clarify and possibly include to the manuscript.

3.      Section 4. It would be better if authors could segregate the molecular level information for functional (e.g. in- lines201 onwards when authors refer to increases in hippocampal baseline excitability, shifts in EEG spectral power,) read out of MIA. Would be better if it is possible to specify what molecular level interplay is going on for a specific observation. Though Fig.2. is rather more generic.

4.      Does fetus age corelate with some of the inflammatory responses or the response to the infection is similar throughout, if this information is available then please review and include to the text

5.      Fig.3 is included twice, not required.

6.      Section 5 lines 325-29, needs further explanation, why males are so prone, please include the specific information if possible.

7.      Authors may consider organizing their manuscript according to the animal model systems mentioned in Fig.1.

8.      Authors may consider another ‘Title’ for their manuscript, highlighting the pathological aspects.

9.      Please improve the quality of the pictures.

Comments on the Quality of English Language

 Please look through the manuscript for typos. Also, a clear flow of ideas and arguments is expected, therefore, please pay careful attention to the sentence structure.

Author Response

Reviewer 2:

  1. Authors are advised to check their references regarding the inflammatory markers.
  • Thank you for pointing this out. We have fixed one reference and added additional references for clarity.
  1. Do authors propose that the inflammatory reaction/mechanisms among the various animal models are similar. Does the distinction among the models made in Fig.1 has no impact? Please clarify and possibly include to the manuscript.
  • All animal models discussed in this review have the same general innate inflammatory mechanisms. However, the extent and length of the inflammatory reaction may differ due to the timing and severity of the given model. Additionally, the inclusion of inflammatory priming can also modulate the inflammatory response. As the exact methodology can vary within each animal system, it is difficult to draw definite conclusions as to these differences.
  1. Section 4. It would be better if authors could segregate the molecular level information for functional (e.g. in- lines201 onwards when authors refer to increases in hippocampal baseline excitability, shifts in EEG spectral power,) read out of MIA. Would be better if it is possible to specify what molecular level interplay is going on for a specific observation. Though Fig.2. is rather more generic.
  • We have removed this (previously lines 201 onward) from this location, as we agree it was missing context for the molecular/cellular changes underlying these functional changes, as well as not fitting well within that paragraph. We have relocated this to lines 308-313 as the molecular perturbations underlying these functional changes are likely modulated by microglia. We have additionally added clarity to the molecular level changes in the innate immune system on lines 219-229. The intracellular signaling cascades and the release of cytokines and chemokines is the hallmark of the adaptive immune response and will take place regardless of the type of immune activation, with some differences between bacterial and viral binding covered in previously figure 2, now figure 1.
  1. Does fetus age corelate with some of the inflammatory responses or the response to the infection is similar throughout, if this information is available then please review and include to the text
  • The fetal immune system, particularly the adaptive immune system, is immature. Therefore, there are some differences in response to infection depending on the gestational timing. However, the innate immune system is largely similar regardless of timepoint. The neurologic systems and processes that are developing at specific timepoints are likely the larger contributing factor to differences in outcomes. We have expanded on these points on lines 255-277.
  1. Fig.3 is included twice, not required.
  • Thank you for catching this error. We have removed the second inclusion of Figure 3.
  1. Section 5 lines 325-29, needs further explanation, why males are so prone, please include the specific information if possible.
  • We have expanded on the findings and proposed mechanisms for male susceptibility to MIA in lines 373-400.
  1. Authors may consider organizing their manuscript according to the animal model systems mentioned in Fig.1.
  • Thank you for this suggestion. What were previously sections 2 (Benefits and limitations of established HIE animal models) and sections 6 (Animal models of HIA can incorporate MIA or inflammation), have been combined into section 5 (Commonly used animal models of HIE and the incorporation of MIA and inflammation). Section 5 now is organized by animal model and discusses the typically used methodology for HIE models followed by how some studies have incorporated MIA or inflammatory priming. This also leads the paper largely with the section of MIA. On lines 496-502 we also added examples of non-Rice-Vannucci rodent models of HIE and have added those to Figure 4 as well. We have additionally expanded on our introduction per a suggestion from reviewer 1.
  1. Authors may consider another ‘Title’ for their manuscript, highlighting the pathological aspects.
  • We elect to keep the current title for the manuscript due to its focus on our central message.
  1. Please improve the quality of the pictures.
  • We have increased the resolution of the figures to 600 DPI and reinserted them into the manuscript with high fidelity settings in Word.

Round 2

Reviewer 2 Report

Comments and Suggestions for Authors

Thanks for addressing the suggesting comments.

- However I would still suggest to change the word 'importance' in the title.

Comments on the Quality of English Language

Please go through the manuscript for typos and simplification of the sentences.

Author Response

Thanks for addressing the suggesting comments.

- However I would still suggest to change the word 'importance' in the title.

We sincerely appreciate the reviewer’s thoughtful comments and constructive suggestions throughout the reviewing process. However, we disagree with removing the word ‘importance’ in the title because the purpose of his perspective review is to present a detailed literature review presenting evidence that maternal inflammation and infection substantially worsen the outcomes of HIE in both humans and animal models. We feel that this perspective is timely because the incorporation of maternal inflammation/infection is common in humans with HIE yet has not been widely adopted within the HIE research community. We feel that we make a persuasive argument for the importance of incorporating maternal inflammation into animal models of HIE in this manuscript and that ‘importance’ should remain included in the title.

However, based on your feedback, we have changed the title to “The importance of including maternal immune activation in animal models of hypoxic ischemic encephalopathy” to present a clearer message to the reader.

Comments on the Quality of English Language:

Please go through the manuscript for typos and simplification of the sentences.

We have carefully reviewed the manuscript and made several changes for clarity and removed punctuation errors. All edits have been highlighted in yellow or listed in comments.